# Integrated Analysis of Proteomics and Metabolomics for Heat Stress in Chinese Holstein Cows

**DOI:** 10.3390/ani15203049

**Published:** 2025-10-20

**Authors:** Xiao Wang, Yinglin Yuan, Fen Pei, Jian Yang, Chenchen Wang, Peng Bao, Xiuxin Zhao, Huiming Liu, Hongding Gao, Minghai Hou, Yundong Gao, Jianbin Li, Dan Hao, Rongling Li

**Affiliations:** 1Institute of Animal Science and Veterinary Medicine, Shandong Academy of Agricultural Sciences, Jinan 250100, China; xiaowangzntc@163.com (X.W.); yuanyl2024@shanghaitech.edu.cn (Y.Y.); houminghai@sdox.cn (M.H.); gaoyundong@sdox.cn (Y.G.); msdljb@163.com (J.L.); 2School of Life Science and Technology, Shanghai Tech University, Shanghai 201210, China; 3Shandong OX Livestock Breeding Co., Ltd., Jinan 250100, China; Peifen9809@163.com (F.P.); jianyang0628@163.com (J.Y.); dawnwang019@gmail.com (C.W.); 18653146630@163.com (P.B.); zhaoxiuxin2003@163.com (X.Z.); 4Center for Quantitative Genetics and Genomics, Aarhus University, 8000 Aarhus, Denmark; hliu@qgg.au.dk; 5Natural Resources Institute Finland (Luke), 31600 Jokioinen, Finland; hongding.gao@luke.fi; 6Jinan Key Laboratory of Poultry Germplasm Resources Innovation and Healthy Breeding, Poultry Institute, Shandong Academy of Agricultural Sciences, Jinan 250100, China

**Keywords:** heat stress, differentially expressed protein, metabolite, protein-metabolite interaction, Chinese Holstein

## Abstract

**Simple Summary:**

Heat stress (HS) severely jeopardizes thermoregulation in dairy cows, significantly reducing productivity. The present study revealed 29 differentially expressed proteins and 338 differential metabolites by using TMT-based proteomes and an untargeted metabolomics approach in the heat-stressed (*n* = 6) and heat-resistant (*n* = 6) groups, respectively. Combined analysis revealed four key pathways underlying protein–metabolite interactions, where up-regulated PLOD1 and ACTN4 and down-regulated EXT1 and GSN interacting with the down-regulated N6-Acetyl-L-lysine, citric acid, 4-Pyridoxic acid, uracil, and uric acid and the up-regulated arachidonic acid were enriched. Interference of the *ACTN4* gene could induce dairy cow mammary epithelial cells apoptosis, which could be regarded as a potential biomarker for HS in Chinese Holstein. These findings provide new insights into the molecular mechanisms underlying HS in Chinese Holstein.

**Abstract:**

Heat stress (HS) severely significantly reduces milk yield and causes substantial economic losses of dairy cows. TMT-based proteomes and an untargeted metabolomics approach were used to conduct the proteomics and metabolomics in heat-stressed (HS, *n* = 6) and heat-resistant (HR, *n* = 6) Chinese Holstein. The proteomics showed that 29 differentially expressed proteins (DEPs), with SERPINA3-7, ACTN4, and PLOD1 up-regulated, and GSN down-regulated in HR cows. The metabolomics showed that 168 differential positive metabolites and 170 differential negative metabolites were identified, with HR cows exhibiting lower levels of anti-inflammatory compounds, such as N6-Acetyl-L-lysine. In addition, 29 DEPs and 338 metabolites revealed four key pathways, including the lysine degradation (ko00310) and metabolic pathway (ko01100) with underlying protein–metabolite interactions, where up-regulated PLOD1 and ACTN4 and down-regulated EXT1 and GSN were observed to be interacting with the down-regulated N6-Acetyl-L-lysine, citric acid, 4-Pyridoxic acid, uracil, and uric acid, and the up-regulated arachidonic acid was enriched, which could be used for rapid and noninvasive screening of heat-tolerant cows. Functional validation through cell experiments, qPCR, and Western blot analyses showed that the interference of the *ACTN4* gene could induce dairy cow mammary epithelial cell apoptosis, which could be regarded as a potential biomarker for HS in Chinese Holstein. Our results facilitate a better understanding of the molecular mechanism underlying the HS issue in dairy cows and provide a crucial insight into the alternative strategies to enhance animal welfare and productivity under high-temperature conditions.

## 1. Introduction

In dairy cattle, heat stress (HS) frequently occurs when they are unable to adequately maintain thermal body balance due to excess endogenous or exogenous heat. This condition severely results in disrupted thermoregulation and subsequent alterations in physiological, hematological, and hormonal functions [1]. HS strongly suppresses feed intake, weakens immunity function, reduces milk yield, impairs reproductive performance, and finally results in significant economic losses [2,3]. Currently, rectal temperature (RT) and respiration rate are widely employed as indicators to evaluate HS in dairy cows [4,5]. However, these traditional indicators are incapable of capturing underlying molecular response, so it is challenging to assist cattle breeders to improve dairy cows’ adaptability to severe HS conditions.

Proteomics can quantify circulating protein abundance in various bovine tissues under HS conditions, such as plasma [6,7,8], liver [9], adipose [10], and mammary epithelial cells [11]. It serves as a high-throughput tool for studying HS-induced physiological alterations in dairy cows. Additionally, blood plasma protein content could serve as a potential biomarker for physiological status indicators, as blood interacts with various tissues during circulation and is relatively easy to obtain in dairy cows [12].

Metabolomics can clarify the small-molecule metabolites for the complete final product of biological regulatory processes. The metabolites could be regarded as biomarkers of physiological alterations from HS condition [6,13]. HS induces metabolic changes to maintain constant body temperature in dairy cows [11]. It potentially leads to the disordered metabolism of induced lipolysis in the adipose tissue [14] and the decreased levels of non-esterified fatty acid and glucose [15]. Furthermore, it reduces milk yield and protein content [14,16], and increases mammary immune response [17]. Since the metabolites are dynamic and influenced by various conditions, including genetics and environments [13,18], it is necessary to explore metabolite changes under HS conditions in dairy cows.

The current omics analysis can provide a better and deeper view into molecular mechanisms in farm animals that may prioritize addressing the HS issue for Chinese Holstein cows [19,20]. Previous researchers studied HS molecular mechanisms in dairy cows at the proteomics or metabolomics level, but few studies integrate them to generate a more comprehensive understanding for interpreting HS regulation mechanisms. Our study comprehensively integrated proteomics and metabolomics techniques to conduct an in-depth analysis for HS mechanisms in Chinese Holstein cows. Cows in one barn were distinguished as being in heat-stressed (HS) or heat-resistant (HR) conditions according to the difference in their physiological responses, i.e., 12 cows with higher (*n* = 6) and lower (*n* = 6) first principal component (PC1) values were selected from 68 cows. The study aims to reveal the molecular mechanisms by which cows respond to HS at the proteomics and metabolomics level, provide a theoretical basis for alleviating the impact of HS condition, and improve the farming efficiency of dairy cows.

## 2. Materials and Methods

### 2.1. Dairy Cows and Sample Collections

In this study, an approximate population size of 1000 lactating Chinese Holstein dairy cows in healthy status were housed in one farm that was equipped with the same fans and sprinklers in Jinan city, China [5]. Since the cows were in the process of maintaining peak milk production in the middle of their first lactation period (122.2 ± 11.6 days), we selected 68 cows with similar age (28.1 ± 0.4 months) and daily milk yield (39.1 ± 1.9 kg per day) during this period. They were all fed with the same total mixed rations and clean water to keep the same environmental conditions. We recorded separately the RT values for five consecutive days from 5th to 9th May and 25th to 29th July in 2020, between 13:00 and 15:00 with a digital thermometer (GLA M800, GLA Agricultural Electronics, San Luis Obispo, CA, USA). The ambient temperature (T) and relative humidity (RH) were also recorded to calculate the temperature–humidity index (THI) values. The THI values were calculated following the formula of THI = (1.8 × T + 32) − [(0.55 − 0.0055 × RH) × (1.8 × T − 26)] in the previous study [5,21]. Based on our previous study, the THI range in the barn was 61–65 in early May which is not in an HS state, but its value was 78–82 in late July which indicates heat stress [5]. In addition, we used principal component analysis (PCA) for the RT and milk yield of these 68 cows. The first principal component (PC1) explained that the 58% variation in the change in RT and milk yield was due to HS. According to PC1, 12 Chinese Holstein cows were selected where 6 cows were assigned to the HS group (PC1 > 1.3) and 6 cows were assigned to the heat-resistant (HR) group (PC1 < −1.2). The average RT and milk yield for 5 days between the two groups exhibited a significant difference in late July [5]. More detailed information for grouping the HS and HR cows using the PC1 values has been carefully described in our previous study [5].

For the plasma samples collected from the tail veins of each of the individual 12 selected cows at the end of July, one 10 mL vacutainer blood collection tube containing the EDTA-coated RNase-free was used. All collected samples were centrifuged at 3000× *g* for 10 min at 4 °C. The 400 µL supernatant of them were immediately transferred to liquid nitrogen for 15 min of freezing.

### 2.2. Proteomics Sequence Procedures

We randomly selected 4 samples from each group to do the proteomics analysis. The most abundant proteins of serum were removed following the manufacturer’s protocol (Agilent Technologies, Santa Clara, CA, USA). We added the SDT Lysis buffer (2% Sodium dodecyl sulfate (SDS), 100 mM Dithiothreitol (DTT), and 100 mM Tris-HCl), boiled for 15 min, and then centrifuged at 14,000× *g* for 40 min. Then, the supernatant was quantified with a bicinchoninic acid assay (BCA) kit (P0012, Beyotime) and stored at −20 °C for further analysis. We selected 20 µg of proteins from each sample and mixed them with 6X loading buffer, boiled for 5 min, and subjected them to a 12% sodium dodecyl sulfate polyacrylamide gel electrophoresis (SDS-PAGE) gel (250 V, 40 min) to test the quality of the extracted proteins by Coomassie Blue R-250 staining (Appendix A).

We employed the filter-aided sample preparation (FASP) procedure for protein digestion [22], with 200 μg of protein and DL-Dithiothreitol (DTT) added to each sample to ensure a final concentration of 100 mM. Then, the samples were boiled for 5 min, centrifuged at 12,500× *g* for 25 min, and transferred to a 30 kDa ultrafiltration centrifuge tube. Subsequently, 100 μL of IAA buffer was added and reacted for 30 min at room temperature in the dark. After alkylation, 100 μL of UA buffer, followed by 0.1 M of triethylammonium bicarbonate (TEAB) solution and 40 μL of trypsin buffer (4 μg trypsin in 40 μL 0.1 M TEAB solution) were added to each sample in a step-by-step manner. The samples were then placed at 37 °C for 16–18 h, followed by centrifugation at 12,500× *g* for 15 min, and the filtrate was collected. Finally, 100 μg of the resulting polypeptide mixture for each sample was labeled using a tandem mass tag (TMT) reagent according to the manufacturer’s instructions (Thermo Fisher Scientific, Sunnyvate, CA, USA).

The TMT-labeled peptides were fractionated by RP chromatography using the Agilent 1260 Infinity II HPLC. Briefly, the chromatographic column was balanced with buffer A (10 mM HCOONH_4_, 5% acetonitrile, pH = 10.0) and buffer B (10 mM HCOONH_4_, 85% acetonitrile, pH = 10.0) for 30–65 min at a flow rate of 1 mL/min with a phase gradient (Appendix A). After the 40 eluted fractions were collected at 214 nm absorbance value during the elution process, the peptides were combined into 10 fractions and dried by vacuum centrifuging at 45 °C.

### 2.3. Mass Spectrometry Analysis for Proteomics and Data Processing

We used a Q Exactive HF-X mass spectrometer (Thermo Fisher Scientific, Sunnyvate, CA, USA) coupled to Easy nLC (Thermo Fisher Scientific Inc.) for the mass spectrometry (MS) analysis for 60 min. MS data was acquired using a data-dependent method to dynamically choose the most abundant precursor ions from the survey scan (350–1800 *m*/*z* value) for HCD fragmentation. Survey scans were acquired at a resolution of 70,000 at *m*/*z* value 200 with an AGC target of 3 × 10^6^ and a max IT of 50 ms. MS2 scans were acquired at a resolution of 17,500 for HCD spectra at *m*/*z* value 200 with an AGC target of 2 × 10^5^ and a max IT of 45 ms with an isolation window of 2 *m*/*z* value. Only ions with a charge state between 2 and 6 and a minimum intensity of 2 × 10^3^ were selected for fragmentation. Dynamic exclusion for the selected ions was 30 s to avoid repeated fragmentation. Normalized collision energy was set as 30 eV.

We used Mascot (version 2.6) and Proteome Discoverer (version 2.2) (Orsburn, 2021) to process the MS/MS raw files for protein identification using the UniProt and NCBInr databases as bovine reference. The precursor mass tolerance was set as 10 ppm and 0.05 Da tolerance for MS2 fragments. The search parameters against databases included trypsin as the enzyme used to generate peptides, with a maximum of 2 missed cleavages permitted. A peptide and protein with a false discovery rate (FDR) of 1% was enforced using a reverse database search strategy. The differentially expressed proteins (DEPs) were identified by the thresholds of fold change > 1.2 and *p*-value < 0.05 based on Student’s *t*-test.

### 2.4. Subcellular Localization, Structural Domain, and Transcription Factor Prediction

The prediction software WoLF PSORT 2017 [23] was used for subcellular localizations of DEPs to convert protein sequences into digital localization features based on sorting signals, amino acid composition, and functional motifs. Subsequently, the subcellular localizations of proteins were predicted using the K-nearest neighbor classifier. The structural domain of a protein is a region with a specific structure and an independent function. Studying the structural domain of a protein is of great significance for understanding the biological function of the protein and its evolution [24]. We used the Interpro database to predict the structural domain of the identified protein [25]. Transcription factor (TF) proteins are proteins that can specifically bind to a specific DNA sequence to regulate gene expression. Therefore, we predicted TFs for DEPs based on the Animal Transcription Factor Database (AnimalTFDB) (version 4.0) [26] that contains protein sequences of 138 animals’ genomes and information on transcription factors.

### 2.5. Protein–Protein Interaction (PPI) Network

We used the protein–protein interaction (PPI) network to understand the signaling pathways and protein complex identification using computational methods in a specific biological process. The STRING database (version 10) [27] provides an integration of protein–protein interaction across more than 2000 organisms, so we used it to examine the interactions among DEPs.

### 2.6. Metabolite Extraction and LC-MS Analysis and Data Processing

We extracted the metabolites with 50% methanol buffer on ice, incubated at room temperature for 10 min, and stored overnight at −20 °C. After centrifugation at 4000× *g* for 20 min, the supernatants were transferred into 96 well plates and then stored at −80 °C before the LC-MS analysis. The LC-MS analysis was performed by TripleTOF 5600 Plus high-resolution tandem mass spectrometer (SCIEX, Warrington, UK) with both positive and negative ion modes. It was introduced for the separation of metabolites, where the mobile phase consisted of solvent A (water, 0.1% formic acid) and solvent B (Acetonitrile, 0.1% formic acid). The gradient elution conditions with a flow rate of 0.4 mL/min were as follows: 5% solvent B for 0–0.5 min; 5–100% solvent B for 0.5–7 min; 100% solvent B for 7–8 min; 100–5% solvent B for 8–8.1 min; and 5% solvent B for 8.1–10 min.

The acquired LC-MS data pretreatment was performed to transfer them into mzXML data format using MSCconvert software (version 3.0). Raw data files were processed by R packages XCMS (version 3.8) [28], CAMERA (version 3.2) [29], and metaX (version 1.4) [30]. XCMS was used to acquire peak values, correct retention time, and annotate isotopes and adducts. CAMERA was used to add and annotate the extracted substances. Metabolites were identified by metaX. The Kyoto Encyclopedia of Genes and Genomes (KEGG) [31] and Bovine Metabolome Database (BMDB) [32] were used to annotate the metabolites. Low-quality peaks of more than 50% missing in QC samples or more than 80% missing in collected samples were removed. The K-nearest neighbors’ method was used to fill the missing values. PCA was performed for the outlier detection and batch effects were corrected to the QC data by using the robust spline correction method. The metabolite pathway was enriched using the MetaboAnalyst software (version 6.0) [33].

### 2.7. Cell Culture and Transfection

To validate the functions of actinin alpha 4 gene (*ACTN*4) on apoptosis in epithelial cells, the dairy cow mammary epithelial cells (DCMECs) were cultured following the previous study [34]. Human embryonic kidney 293T (HEK 293T) cells were purchased from the American Type Culture Collection (ATCC) and cultured following the previous study [35]. DMECs were cultured at 37 °C in the presence of 5% CO_2_ and maintained in DMEM/F12 with 10% FBS (Servicebio, Wuhan, China) and antibiotics (100 µg/mL streptomycin and 100 U/mL penicillin). The cells were transfected with three siRNAs using Lipofectamine 2000, and the three siRNAs were synthesized by the company (Shanghai GeneChem Co., Ltd, Shanghai, China) listed in Appendix A. Cell apoptosis was measured using the Annexin V-FITC/PI Apoptosis Kit (AT101) according to instructions (Multisciences (Lianke) Biotech Co., Ltd., Hangzhou, China).

### 2.8. Quantitative Reverse Transcription Polymerase Chain Reaction

Total RNA was extracted using TRIzol reagent (Tiangen Biotech; China; DP424) and cDNA was synthesized using the FastKing RT Kit (Tiangen Biotech; China; KR116). The mRNA expression levels of *ACTN*4, heat shock protein family A member 1A gene (*HSP70*), heat shock protein 27 gene (*HSP27*), caspase3 gene (*CASP3*), cytochrome c somatic gene (*cytc*), and BCL2 apoptosis regulator gene (*bcl*-2) were analyzed via qPCR with SuperReal PreMix Color (SYBR Green) (Tiangen Co., Beijing, China) and specific primers (Appendix A).

### 2.9. Western Blot

We extracted 1 mL of blood samples and added 100 μL of RIPA lysis buffer for lysis for 30 min. Protein expression of heat shock protein 105 kDa (HSPH1) and lipopolysaccharide-binding protein (LBP) were evaluated by Western blot according to the previously established protocols [36]. HSPH1 Antibody (66723-1-IG, Invitrogen) at dilution of 1:10,000 and LBP Antibody (RR433-8, Invitrogen) at dilution of 1:50 were incubated at room temperature for 1.5 h. The ImageJ software (version 1.49, Java 22) was used to calculate the integrated density for the band, and each band was measured three times and *t*-test was used to calculate the significant difference between two groups.

### 2.10. Bioinformatics Analysis

Based on Fisher’s exact test and topology measure with degree centrality, the adjusted *p*-values of gene ontology (GO) and KEGG enrichment for DEPs were set to 0.05 for multiple tests using an FDR Benjamini–Hochberg method. Partial least squares-discriminant analysis (PLS-DA) was conducted via metaX to discriminate the different variables between groups. A variable important for the projection (*VIP*) cut-off value of 1.0 was used to select important metabolites. The integrated pathway analysis of DEPs and the differential metabolites was performed by the MetaboAnalyst (version 6.0) software based on the combined *p*-values at the pathway level [33]. In the pathway enrichment analysis, *Bos taurus* was used as the library. *Fisher’s* exact test for over-representation analysis was implemented to evaluate whether a particular metabolite set is over-represented using the hypergeometric test. In each pathway, pathway impact was calculated as the sum of importance measures of the matched metabolites divided by the sum of the importance measures of all the metabolites. Relative betweenness centrality was selected for pathway impact value calculation and the node importance measure in the topological analysis.

## 3. Results

### 3.1. Protein Quality, Characterization, and Differentially Expressed Protein

Quantification and SDS-PAGE results showed that the protein concentration ranges from 4.4 to 6.3 μg/μL and the total amount ranges from 880 to 1260 μg (Appendix A), indicating a good protein evaluation for sequencing (Appendix A and Appendix A). Peptide mass deviation showed that more than 99.5% of the peptide quality deviation was within 8 ppm (Appendix A), indicating that the instrument state was in a good condition and the mass spectrometry results were credible. We also found the cumulative percentage of peptides in a stable status, even if the corresponding ion score was higher than 90 (Appendix A). The relative protein molecular mass distribution indicated a larger percentage of molecular weight of peptides at 10–100 kDa (Appendix A), while the length of the peptides in amino acids which equaled to 8–9 occupied the highest number (>1000) (Appendix A). The protein sequence coverage distribution indicates that most of the protein sequence coverage was more than 5% (Appendix A). In addition, the PCA cluster was distinct (Figure 1A), and the relative standard deviation (RSD) was smaller (Figure 1B) between the HS and HR groups. Pearson correlation coefficient results (Figure 1C) indicated the weak relationships among the 12 samples.

In total, we matched 34,925 spectra from 385,242 identified spectra, where 4767 of them were unique peptides. The identified protein number was 624 which contained at least one unique peptide segment (Figure 1D). According to the thresholds of |Log2(fold change)| > 1.2 and adjusted-*p*-value < 0.05, twenty-nine DEPs between HR and HS were identified, including twenty-one up-regulated and eight down-regulated proteins (Figure 1E, Table 1 and Appendix A). Of note, BOLA, F1MH40, GSN, EXT1, Q2KIX7, THSD7A, Q2KIT0, and CTSS were significantly highly expressed in the HS group, while MMP19, RCN2, ACTN4, PLOD1, SERPINA3-3, SERPINA3-5, and SERPINA3-7 proteins were significantly highly expressed in the HR group (Appendix A). The heatmap showed good hierarchical clusters for HR and HS groups (Figure 1F). In addition, Western blot results of HSPH1 (Appendix A) and LBP (Appendix A) were higher expressed in the HR group than the HS group, which are consistent with the sequence results.

### 3.2. Proteins Subcellular Localization and Structural Domain

Subcellular localization in an automated and high-throughput fashion provides an appealing complement to experimental techniques. The subcellular localization results in the largest number of proteins present in the extracellular space, accounting for 58.6% of the 624 proteins followed by 10.3% of proteins in the cytosol and nucleus (Figure 2A). The structural domain of proteins showed that alpha-1-antitrypsin-like and leucine-rich repeat N-terminal domains were the most enriched structure domain, with enrich factors of 9.78 and 9.22, respectively (Figure 2B).

### 3.3. Functional Enrichment Analysis

Functional enrichment results of DEPs revealed 64 significant GO terms (Appendix A) and 32 KEGG pathways (Appendix A). The top GO term was chromaffin granule (GO:0042583) in the biological process, DNA binding (GO:0003677) in the molecular function, and very-low-density lipoprotein particle assemble (GO:0034379) in the cellular component (Figure 3A). The significant enrichment was the viral carcinogenesis (*p*-value < 0.05), where the alpha-actinin-4 protein encoded by *ACTN4* and lg-like domain-containing protein and gelsolin protein encoded by *GSN* were enriched. Notably, the procollagen-lysine, 2-oxoglutarate 5-dioxygenase 1 protein encoded by *PLOD1* was enriched in the lysine degradation pathway (ko05203) (*p*-value = 0.091). The other DEPs were enriched in glycosaminoglycan biosynthesis-heparan sulfate/heparin (ko00534) (adjusted-*p*-value = 0.091), porphyrin metabolism (ko00860) (*p*-value = 0.133), and mineral absorption (ko04978) (*p*-value = 0.133) (Figure 3B and Appendix A).

### 3.4. PPI Network Analysis and Transcription Factors

The 624 proteins were used to construct the core network, and the combined score of more than 750 was filtered out. The PPI network showed that four DEPs were identified to have connections with at least four proteins (Figure 4A). Heat shock protein 105 kDa encoded by *HSPH1* was connected with four identified proteins, including HSPA8, HSP90AA1, HSPA1A, and TRAP1. Lumican, encoded by *LUM*, is connected with 12 proteins, including B4GALT1, B3GNT1, COL1A1, COL1A2, COL3A1, COL6A3, COL6A1, FMOD, FN1, HSPG2, MMP2, and POSTN. The protein disulfide-isomerase encoded by *P4HB* is connected with eight proteins, including APOB, CALR, heat shock 70 kDa protein 1A (HSPA1A), HYOU1, PRDX4, PPIB, protein disulfide isomerase family A member 3 (PDIA3), and TXN. The gelsolin encoded by *GSN* is connected with five proteins, including ACTA2, ACTG1, FN1, TMOD2, and VCL.

Three DEPs in the SERPINA3 family are involved in maintaining intracellular homeostasis. The PPI network showed that the key proteins of hemopexin (HPX), albumin (ALB), glycocalincin (GC), and alpha-1-β glycoprotein (A1BG) were identified in the network (Figure 4B). The HSPH1 and P4HB proteins were significantly enriched in the protein processing in endoplasmic reticulum pathway (ko04141) and the PPI network for the proteins enriched in the protein processing in endoplasmic reticulum pathway (Figure 4C). The DEPs were annotated in five TF families and they were the CLS TF of the Ig-like domain-containing protein (LOC534578) and VCAM1 protein, the ZBTB TF of the Kelch-like family member 8 and Galectin-3-binding protein, the HSF-DNA-binding TF of the heat shock transcription factor 4 (HSTF4) protein, the RXR-like TF of the nuclear receptor subfamily 2 group protein, and the THR-like TF of the peroxisome proliferator-activated receptor delta (PPARD) protein (Figure 4D).

### 3.5. Metabolite Identification

We finally identified 21,008 ion features and their total ion chromatogram (TIC) distribution was listed to reflect the retention time of all the samples in both positive and negative metabolites (Appendix A). The *m*/*z*-rt distribution of positive and negative metabolites showed the highest metabolite feature number when the average retention time was from 2 to 3.5 min and the average *m*/*z* range was from 200 to 400 (Appendix A). The highest number of identified metabolites in positive and negative ions were lipids and lipid-like molecules, phenylpropanoids and polyketides, and organ heterocyclic compounds, based on HMDB super class classification (Figure 5A and Figure 6A). PCA results showed that all QC samples were clustered together to confirm the stable and reliable assay (Figure 5B and Figure 6B).

The PC1 and PC2 of PLS-DA explained 10.01% and 11.07% variations in all positive metabolites (Figure 5B) and explained 9.72% and 10.25% variations in all negative metabolites (Figure 6B). The permutation results confirmed that PLS-DA had a *p*-value < 0.05 after 1000 permutation tests in both positive and negative metabolites. In addition, the model was well-fitted and highly predictable for our data with R^2^ = 0.993 and Q^2^ = 0.545 in the positive metabolites (Figure 5C), and R^2^ = 0.995 and Q^2^ = 0.578 in the negative metabolites (Figure 6C).

### 3.6. Differential Positive and Negative Metabolites

The number of positive metabolites was 6890, where 100 differential metabolites were up-regulated and 68 differential metabolites were down-regulated (*VIP* >1 and *p* < 0.05) (Figure 5D). The top five of them were trigonelline and phenytoin in the up-regulated status, but 2-methyl-2-(4-methylpent-3-en-1-yl)-2H-chromen-8-ol, atorvastatin, and phaeophorbide b were in the down-regulated positive status (Table 2). The heatmap exhibited the differential positive metabolites clustered in the different groups (Figure 5E). The enrichment analysis of differential positive metabolites found glycerophospholipid metabolism (map00564), porphyrin metabolism (map01100) (biliverdin and bilirubin), and sphingolipid metabolism (map00600) (phosphatidylethanolamine, phosphatidylcholine and phosphatidylserine) as the top three pathways (Figure 5F).

The number of negative metabolites was 7011, whereby 85 differential metabolites were up-regulated and 85 differential metabolites were down-regulated (*VIP* > 1 and *p*-value < 0.05) (Figure 6D). The top five of them were valganciclovir and 5-Methylcytidine in the up-regulated negative status, but MG(0:0/20:1(11Z)/0:0), penicillin G, and 6-[(2-carboxyacetyl)oxy]-3,4,5-trihydroxyoxane-2-carboxylic acid were in the down-regulated positive status (Table 2). The different heatmap clusters were also observed in the differential negative metabolites in the other groups (Figure 6E). The drug metabolism-cytochrome P450 (map00982), glycerophospholipid metabolism (map00564), and glycerolipid metabolism (map00561) were the top three enriched pathways (Figure 6F).

### 3.7. Integrated Analysis for Proteomics and Metabolomics

To explore the protein–metabolite interactions, we revealed four pathways from the 32 significantly enriched pathways of the 29 DEPs and 39 significantly enriched pathways of the 338 metabolites (Figure 3, Figure 5 and Figure 6; Appendix A and Appendix A). The four pathways were the lysine degradation, the metabolic pathway, Fc gamma R-mediated phagocytosis, and amoebiasis (Table 3). The up-regulated protein of PLOD1 and the down-regulated protein of EXT1 were involved in regulating metabolites like the up-regulated arachidonate, the down-regulated citrate, the down-regulated uracil, and the down-regulated uric acid through the metabolic pathway. The binding of N6-Acetyl-L-lysine metabolites with PLOD1 proteins was identified through the lysine degradation pathway. Amoebiasis pathways contain ACTN4 protein and arachidonic acid metabolite.

### 3.8. Interference of ACTN4 Promotes Apoptosis in Epithelial Cells

The relative mRNA expression level of the *ACTN4* gene was higher in HS cows based on qRT-PCR, which is consistent with the proteomics result by TMT (Figure 7A). After transferring the designed shRNAs of *ACTN4* to cells, its expression significantly decreased, and sh-1 showed the best interference effect (Figure 7B). Then, we used sh-1 to interfere with *ACTN4* expression in dairy cow mammary epithelial cells (DCMECs). The early, late, and total apoptosis rates were significantly increased compared with the control group (Figure 7C). We also found that the heat-resistant-related gene *HSP70* (Figure 8A), as well as the cell-apoptosis-related genes *Bax* (Figure 8B), *caspase*-3 (Figure 8C), and *cytc* (Figure 8D) were significantly up-regulated after interfering with the expression of *ACTN4*, but *HSP27* (Figure 8E) and *Bcl-2* (Figure 8F) were not significant.

## 4. Discussion

In this study, the plasma metabolome and proteome were integrated to uncover the molecular regulation of HS to physiology, production, and health in Chinese Holstein dairy cows. Plasma proteomics and metabolomics between HS and HR cows identified 29 proteins and 338 metabolites in the differentially expressed conditions.

Four cytoplasmic vesicle-localized serpin proteins were significantly up-regulated in the HR group, including serpin A3-3, serpin A3-5, serpin A3-7, and serpin A3-8, which modulate the immune responses by targeting proteases involved in HS regulation [37,38,39]. In addition, posits heat shock protein (HSPH1), divergently expressed in indigenous and crossbreed cattle [37], was up-regulated in the HS group, which was consistent with previous studies that in the blood of Zebu cattle [40] and the liver [41] of lactating dairy cows after heat exposure. We also found that HSPH1 protein is connected with HSPA8 [42] and HSP90B1 [41], which are critical for coordinating the changes in the biological system under HS condition.

The differential metabolites of N6-Acetyl-L-lysine, citric acid, 4-pyridoxic acid, uracil, uric acid, and arachidonic acid interacting with the four DEPs were involved in human diseases like primary glomerulopathies, COVID-19, Alzheimer’s disease, obesity, and metabolic dysfunction-associated steatotic liver diseases [43,44,45,46]. Nα-acetyl-α-lysine plays a role in HS responses in various organisms. It can accumulate and act as a probable thermolyte to alleviate HS after increasing in concentration under elevated temperatures [47]. The application of citric acid may alleviate growth and physiological damage caused by high temperature, as exogenous citric acid enables the maintenance of membrane stability and root activity. Meanwhile, citric acid can activate HSP genes and the antioxidant response to protect tall fescue against HS-induced damage in the plant study [48]. As a metabolite of vitamin B6, 4-Pyridoxic acid has been observed to increase under HS conditions, together with inflammatory and oxidative stress in various studies. It suggested the disruptive role of HS on the body’s normal vitamin B6 metabolism [49]. As an intermediate breakdown product of guanosine monophosphate (GMP) and uracil, dihydrouracil is decreased, probably because of the defects of human metabolism. Of note, GMP production is mediated by CD39 and CD73 and their axis plays a crucial role in immunity and inflammation [50,51]. Uric acid is generated during HS partly due to nucleotide release from muscles. HS and high uric acid levels are linked, especially when the individuals are exposed to the heating conditions with inadequate hydration [52].

Four DEPs (i.e., up-regulated PLOD1 and ACTN4 and down-regulated EXT1 and GSN) interacting with six differential metabolites (i.e., the down-regulated N6-Acetyl-L-lysine, citric acid, 4-Pyridoxic acid, uracil, uric acid, and the up-regulated arachidonic acid) were enriched in four key pathways of underlying protein–metabolite interactions. The lysine degradation was where PLOD1-N6-Acetyl-L-lysine interaction enriched (Table 3). PLOD1 plays an essential role in protein synthesis, folding and trafficking, as well as participating in fatty acid metabolism and collagen synthesis [53,54]. In addition, PLOD1 can enhance the transcriptional activity of heat shock factor 1 (HSF1) [55]. Hence, HSF1 is activated during thermal stress and may be caused by the transcriptional activation of PLOD1 [55]. More interestingly, the amoebiasis was where ACTN4-arachidonic acid interaction enriched. ACTN4 protein is an important structural protein for cytoskeleton organization and the maintenance of cellular integrity, which also plays an important role in adipogenesis and marbling deposition in cattle [56,57]. The release of arachidonic acid increased after HS, triggering autophagy in sertoli cells via dysfunctional mitochondrial respiratory chain function, which potentially affects cellular function and reproductive health [58]. In addition, previous researchers also found that arachidonic acid had high sensitivity and specificity in diagnosing HS status [59], which could be regarded as one of the biomarkers for HS in dairy cows. However, the molecular relationship between PLOD1 and HSF1 or ACTN4 and arachidonic acid to HS needs more experimental verification in dairy cows.

## 5. Conclusions

In summary, our study identified 29 differentially expressed proteins (DEPs) and 338 differential metabolites after comparing HS and HR conditions. The integrated analysis of 29 DEPs and 338 metabolites revealed four key pathways underlying protein–metabolite interactions, where PLOD1, EXT1, GSN, and ACTN4 interacted with N6-Acetyl-L-lysine, citric acid, 4-Pyridoxic acid, uracil, uric acid, and arachidonic acid. Functional validation through cell experiments, qPCR, and Western blot analyses showed that interference of *ACTN4* gene could induce dairy cow mammary epithelial cell apoptosis, which could be regarded as a potential biomarker for precision selection of heat-tolerant cows. Overall, these findings enhanced our understanding of molecular mechanisms at proteomics and metabolomic levels by which dairy cows respond to HS condition, facilitating novel insights that may inform strategies to mitigate stress using potential applications, e.g., targeted breeding strategies or nutritional interventions, and enhancing animal welfare and productivity under high-temperature conditions.

## Figures and Tables

**Figure 1 animals-15-03049-f001:**
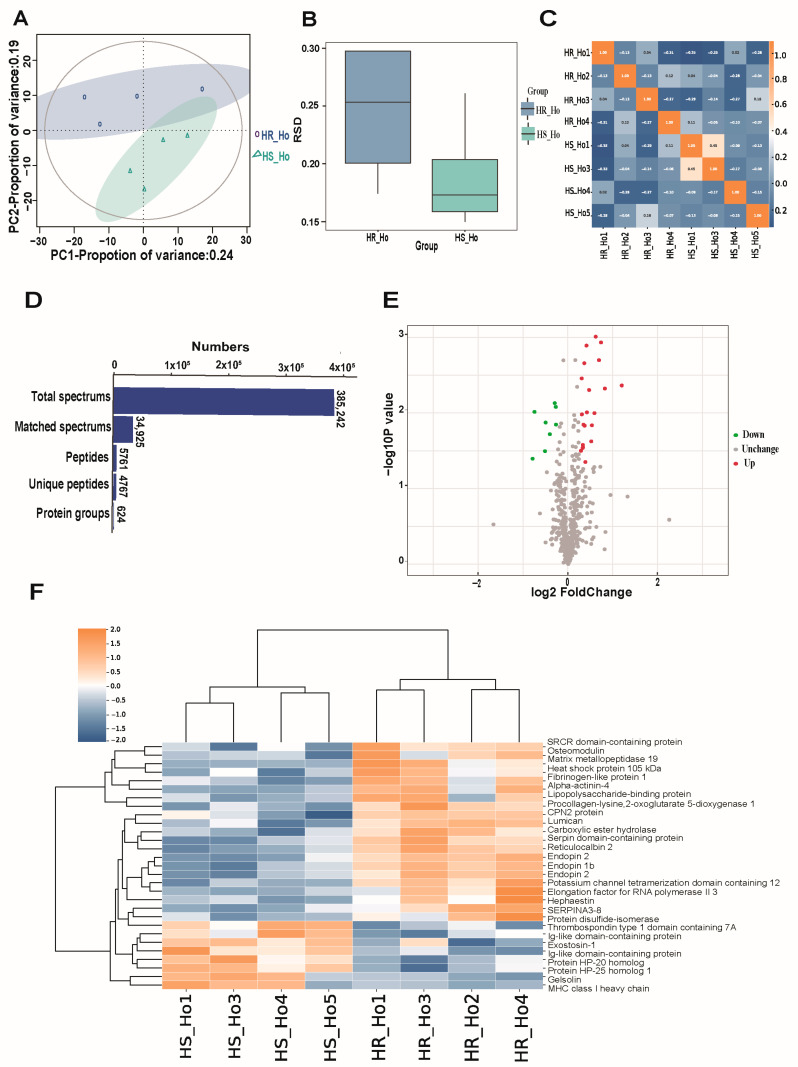
Comparison analysis of protein expression levels between heat-stressed (HS) and heat-resistant (HR) cows. (**A**) Principal component analysis (PCA) of proteomics data between HR and HS groups. (**B**) Relative standard deviation analysis of quantitative protein values. (**C**) The heatmap of Pearson correlation coefficients among HR and HS groups. (**D**) Statistical summary for protein identification and classification. (**E**) Volcano plot for the identified proteins. The *X*-axis indicated the Log2(Fold change) value and the *Y*-axis indicated the −log_10_(*p*-value). (**F**) The heatmap of the differentially expressed proteins (DEPs) between HR and HS groups.

**Figure 2 animals-15-03049-f002:**
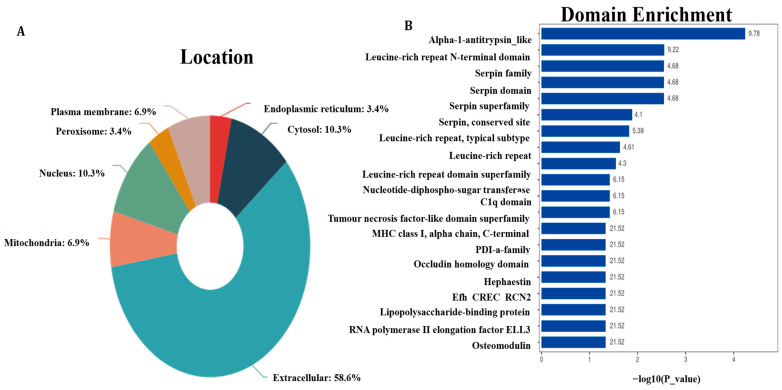
Protein subcellular localization and structural domain. (**A**) Distribution of protein subcellular localization. (**B**) Histogram of enrichment statistics for the top 20 protein structural domain classification.

**Figure 3 animals-15-03049-f003:**
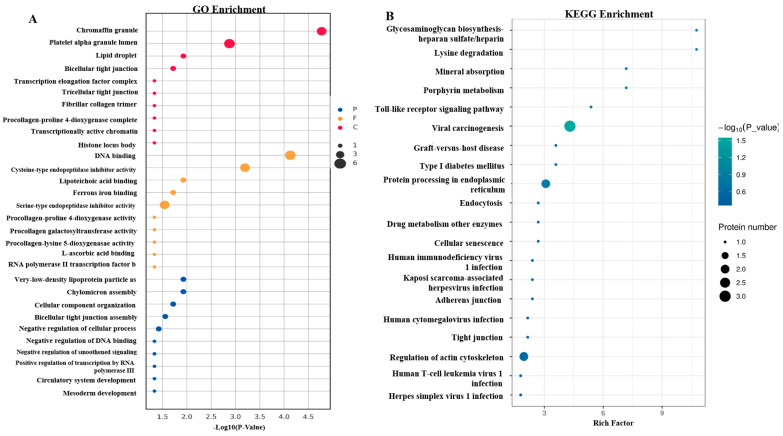
Functional enrichment results of GO terms and pathways of differentially expressed proteins (DEPs). (**A**) Histogram of enrichment statistics for the top 10 GO terms. (**B**) Bubble plot of enrichment statistics for the top 20 KEGG pathway. The vertical coordinate indicates the KEGG; the horizontal coordinate indicates the rich factor value; the green color indicates the smaller *p*-value; and the circle size indicates the number of DEPs contained in the pathway.

**Figure 4 animals-15-03049-f004:**
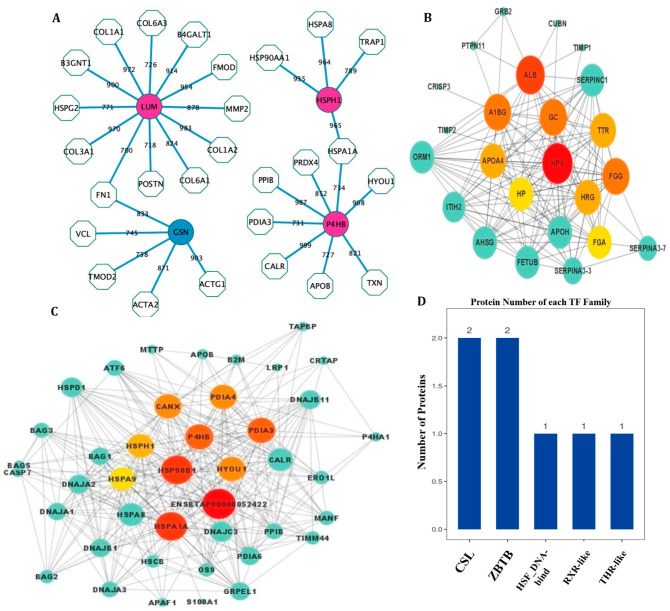
Protein–protein interaction (PPI) network and transcription factor results. (**A**) The PPI network results. Octagon indicates the identified proteins. Ellipse indicates the differentially expressed proteins (DEPs). The combined scores were shown on each edge. The red color represents the proteins highly expressed in the heat-stress group; the green color represents the proteins highly expressed in the heat-resistant group. (**B**) PPI based on the SERPINA3 family. (**C**) PPI based on HSPH1 and P4HB. (**D**) The number in each transcription factor family.

**Figure 5 animals-15-03049-f005:**
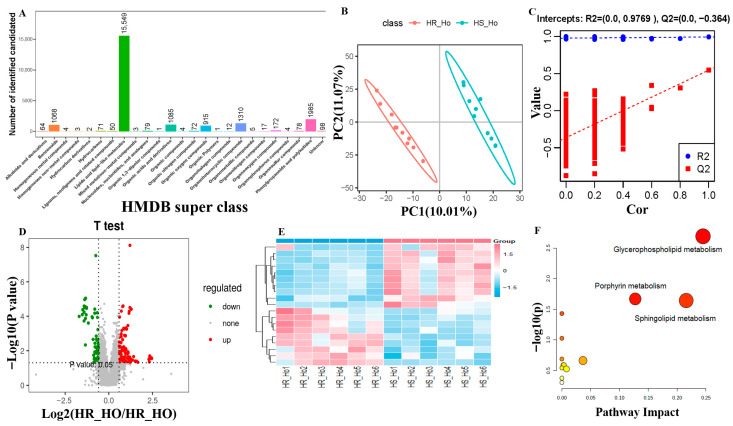
Comparison analysis of positive metabolite abundance levels between heat-stressed (HS) (n = 6) and heat-resistant (HR) cows (n = 6). (**A**) Positive metabolite classification based on the Human Metabolome Database (HMDB). (**B**) The partial least squares-discriminant analysis (PLS-DA). (**C**) Permutation test diagram. (**D**) Volcano plot for the identified metabolites. The horizontal coordinate is the difference multiple of metabolic ions in the comparison group. The ordinate is the *p*-value. (**E**) The heatmap of differentially expressed metabolite levels between HR and HS groups. (**F**) Differential metabolite pathway analysis.

**Figure 6 animals-15-03049-f006:**
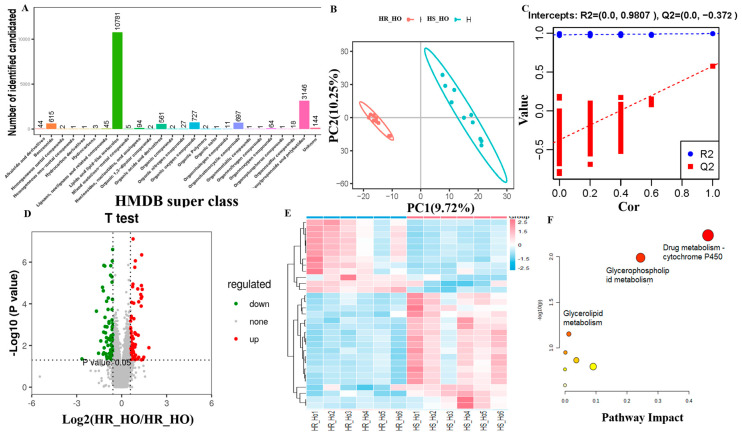
Comparison analysis of negative metabolite abundance levels between heat-stressed (HS) (n = 6) and heat-resistant (HR) cows (n = 6). (**A**) Negative metabolite classification based on the Human Metabolome Database (HMDB). (**B**) The partial least squares-discriminant analysis (PLS-DA). (**C**) Permutation test diagram. (**D**) Volcano plot for the identified metabolites. The horizontal coordinate is the difference multiple of metabolic ions in the comparison group. The ordinate is the *p*-value. (**E**) The heatmap of differentially expressed metabolite levels between HR and HS groups. (**F**) Differential metabolite pathway analysis.

**Figure 7 animals-15-03049-f007:**
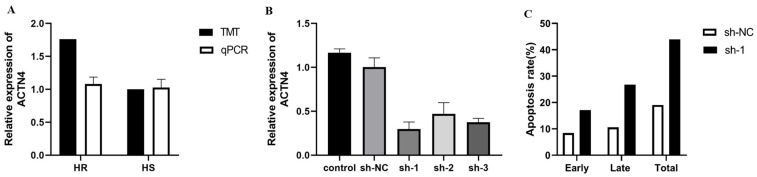
The qRT-PCR validation of functions of *ACTN4* on apoptosis in epithelial cells. (**A**) Relative expression of *ACTN4* between HR and HS cows. (**B**) Relative expression of *ACTN4* after transferring the designed shRNAs of *ACTN4* to cells with interference effect. (**C**) The interference of sh-1 with *ACTN4* expression in the early, late, and total apoptosis rates.

**Figure 8 animals-15-03049-f008:**
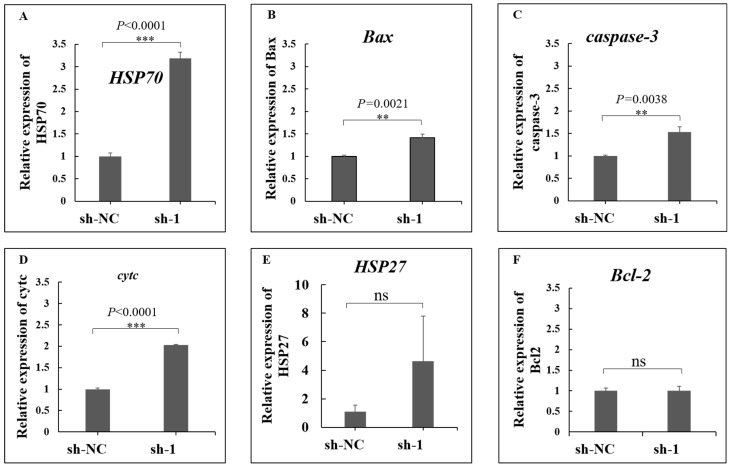
The relative expression of (**A**) heat shock protein family A member 1A gene (*HSP70*), (**B**) Bcl-2-associated X-protein (Bax), (**C**) caspase3 gene (*caspase-3*), (**D**) cytochrome c somatic gene (*cytc*), (**E**) heat shock protein 27 gene (*HSP27*), and (**F**) BCL2 apoptosis regulator gene (*Bcl*-2). Results are shown as mean ± SEM. ns *p* > 0.05, ** *p* < 0.01, and *** *p* < 0.001.

**Table 1 animals-15-03049-t001:** The top 10 differentially expressed proteins (DEPs).

UniProKB	GeneName	AAs	Coverage [%]	Peptides	PSMs	Unique Peptides	AAs	MW	calc.	Score	Regulation	*p*-Value	HR_HO/HS_HO
								[kDa]	pI	Mascot			
A2I7N3	SERPINA3-7	417	49	22	394	13	417	46.9	6.01	6896	UP	0.0043325	2.29
A5D7D1	ACTN4	1032	1	1	1	1	1032	117	6.38	0	UP	0.0047489	1.76
Q3ZEJ6	SERPINA3-3	411	43	15	326	5	411	46.1	6.48	8029	UP	0.0011647	1.66
A2I7N1	SERPINA3-7	415	33	16	448	1	415	46.5	5.81	10,012	UP	0.0020006	1.61
Q0VCQ9	RCN2	317	3	1	1	1	317	36.9	4.4	0	UP	0.0009764	1.53
F6PYF1	MMP19	499	3	1	1	1	499	56.4	6.77	42	UP	0.0100722	1.50
UPI0000616416	MGC137014	191	66	10	113	10	191	20.6	8.59	2185	DOWN	0.0142715	0.83
A0A4W2EXA9	THSD7A	1676	1	1	2	1	1676	186.8	7.43	0	DOWN	0.0083202	0.83
UPI0000693DEA		212	50	9	67	9	212	22.5	7.46	1662	DOWN	0.0074206	0.81
A5D7I4	EXT1	746	2	1	1	1	746	86.2	9.04	30	DOWN	0.0191447	0.75
F1N1I6	GSN	846	61	42	316	1	846	92	6.86	11,356	DOWN	0.0134129	0.71

Note: *p*-values are adjusted. Each group has four samples.

**Table 2 animals-15-03049-t002:** The top five positive and top five negative differential metabolites.

ID	Metabolites	Iron	*p*-Value	*VIP* Value	Up/Down-Regulated Status
M155T113	Trigonelline	Positive	7.57 × 10^−9^	4.00	Up
M283T467	2-methyl-2-(4-methylpent-3-en-1-yl)-2H-chromen-8-ol	Positive	9.03 × 10^−6^	4.07	Down
M581T467	Atorvastatin	Positive	1.08 × 10^−5^	4.16	Down
M607T467	Phaeophorbide b	Positive	2.57 × 10^−5^	4.12	Down
M275T138	Phenytoin	Positive	2.73 × 10^−5^	3.03	Up
M212T138	Valganciclovir	Negative	7.73 × 10^−8^	3.23	Up
M256T65	5-Methylcytidine	Negative	8.66 × 10^−7^	3.45	Up
M383T380	MG(0:0/20:1(11Z)/0:0)	Negative	8.90 × 10^−7^	2.79	Down
M333T96	Penicillin G	Negative	1.48 × 10^−6^	3.02	Down
M279T95	6-[(2-carboxyacetyl)oxy]-3,4,5-trihydroxyoxane-2-carboxylic acid	Negative	4.26 × 10^−6^	3.04	Down

**Table 3 animals-15-03049-t003:** The four key pathways of the integrated analysis of proteomics and metabolomics.

Pathways	Pathway Code	Protein-Coding Gene	Metabolites
Lysine degradation	ko00310	Up-regulated *PLOD1*	Down-regulated N6-Acetyl-L-lysine
Metabolic pathways	ko01100	Up-regulated *PLOD1* and down-regulated *EXT1*	Down-regulated citric acid and down-regulated 4-Pyridoxic acid
			Down-regulated uracil, down-regulated uric acid, down-regulated N6-Acetyl-L-lysine, and up-regulated arachidonic acid
Fc gamma R-mediated Phagocytosis	ko04666	Down-regulated *GSN*	Up-regulated arachidonic acid
Amoebiasis	ko05146	Up-regulated *ACTN4*	Up-regulated arachidonic acid

## Data Availability

The raw omics data used in the current study are available from the corresponding authors.

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
