# Peer review of "Integrated Analysis of Proteomics and Metabolomics for Heat Stress in Chinese Holstein Cows"

_animals, 2025, doi:10.3390/ani15203049_

Round 1

Reviewer 1 Report

Comments and Suggestions for Authors

The subject of this manuscript is highly relevant, timely, and in strong demand, as heat stress in dairy cattle is a critical issue for both production and animal welfare worldwide. The integration of proteomics and metabolomics is valuable and innovative, and the manuscript is overall well written and clearly structured. However, there are several areas that require clarification and improvement to strengthen the paper.

Major Comments

  1. Materials and Methods

    • While the extraction and protein analysis are well described, there is insufficient information regarding the cows themselves.

    • Please clarify:

      • How were the cows selected?

      • What criteria were used to classify them as being under heat stress?

      • For how long were the animals exposed to heat stress conditions?

    • This information is essential for reproducibility and proper interpretation of results.

  2. Lines 83–84 The phrase “first lactating period per lactation” is confusing. Please rephrase to clearly indicate whether you mean first lactation or a specific period within the current lactation.

  3. Lines 84–96: It is unclear how 68 cows and then 12 cows were used. Were 68 cows initially evaluated, and then a subset of 12 was used for the omics analyses?Please clarify the selection process and final sample size used in each analysis step.

Results

  1. Figures

    • The figures are difficult to read due to resolution and size. Please improve the quality and adjust formatting to make results more legible.

  2. Supplementary Data

    • One of the supplementary files includes p-values greater than 1.00, which is not possible. Please check for typographical errors and correct them.

Discussion

  1. Integration of findings. Currently, the proteins and metabolites are discussed separately, without a clear link to their biological implications. Please expand the discussion to explain how these molecular changes affect the physiology, production, and health of cows under heat stress. This would help connect the omics results to practical outcomes and enhance the relevance for dairy science and animal management.

Conclusion

  1. Concluding remarks

    • The conclusion is adequate but could be strengthened by explicitly stating how the integrated proteomic–metabolomic approach contributes to better understanding of heat stress and its potential application in managing dairy cow health and productivity.

Author Response

Dear Reviewers:

We are really appreciating for your comments and suggestions for our manuscript entitled “Integrated analysis of proteomics and metabolomics for heat stress in Chinese Holstein cows (animals-3878666)”. Those comments are all very valuable and very helpful for revising and improving our manuscript. We have carefully considered the comments and made the corrections, which we hope will meet with your approval. The details of the revised sections are marked in red and highlighted in yellow in the manuscript-marked. The main corrections in the paper and the responds to your comments point by point are as flowing:

The subject of this manuscript is highly relevant, timely, and in strong demand, as heat stress in dairy cattle is a critical issue for both production and animal welfare worldwide. The integration of proteomics and metabolomics is valuable and innovative, and the manuscript is overall well written and clearly structured. However, there are several areas that require clarification and improvement to strengthen the paper.

Major Comments

Materials and Methods

While the extraction and protein analysis are well described, there is insufficient information regarding the cows themselves.

  1. Please clarify:
  • How were the cows selected?

Thank you for your question. We have detailed how the cows were selected in our revised manuscript, please see in Line 96-117. Firstly, 1000 lactating Chinese Holstein dairy cows in healthy status were housed in one farm that is equipped with the same fans and sprinklers in Jinan city, China. Subsequently, 68 cows in the middle of their first lactation period (122.2 ± 11.6 days) with similar age (28.1±0.4 months) and daily milk yield (39.1 ± 1.9 kg per day) were selected. They were all fed with the same total mixed rations and clean water to keep the same environmental conditions. Next, we recorded the RT values of the 68 cows for five consecutive days between 13:00 and 15:00 with a digital thermometer (GLA M800, GLA Agricultural Electronics, San Luis Obispo, CA) from May to July in 2020. The ambient temperature (T) and relative humidity (RH) were also recorded to calculate the temperature-humidity index (THI) values. The THI values were calculated following the formula of THI = (1.8 × T + 32) - [(0.55 - 0.0055 × RH) × (1.8 × T - 26)] in the previous study. Based on our previous study, the THI range in the barn was 61-65 in early May which is not in a HS state, but its value is 78-82 in late July which was under heat stress. In addition, we used principal component analysis (PCA) for the RT and milk yield of these 68 cows. The first principal component (PC1) explained the 58% variation of the change in RT and milk yield due to HS. According to PC1, 12 Chinese Holstein cows were selected where 6 cows were assigned to the HS group (PC1 > 1.3) and 6 cows were assigned to the heat resistance (HR) group (PC1 < -1.2). The average RT and milk yield for 5 days between the two groups exhibited a significant difference in late July. More detailed information for grouping the HS and HR cows using the PC1 values has been carefully described in our previous study, “Li, R. et al. (2023) Identification of target genes and pathways related to heat tolerance in Chinese Holstein cows. Livestock Science 271”

  • What criteria were used to classify them as being under heat stress?

Response: Thank you for your question. The criteria of THI were used to classify them as being under heat stress. In our study, the THI value in the barn was 61-65 in early May which is not in a HS state, but its value is 78-82 in late July which was under heat stress. Based on previous studies, THI exceed 72 as a starting point for milk production to decline [1]. Each unit decrease in THI results in a decrease of approximately 0.6 kg in daily milk production after the THI reach 72 [2]. Please see in Line 108-110.

Based on our previous study, the THI range in the barn was 61-65 in early May which is not in a HS state, but its value is 78-82 in late July which was under heat stress [5]

  1. Yousef, M.K. (1985) Stress physiology in livestock. Volume I. Basic principles.
  2. Bohmanova, J. et al. (2007) Temperature-humidity indices as indicators of milk production losses due to heat stress. Journal of dairy science 90 (4), 1947-1956.

  • For how long were the animals exposed to heat stress conditions? This information is essential for reproducibility and proper interpretation of results.

Response: Thank you for your question. We separately recorded the RT values for five consecutive days in 5st to 9st in May and 25st to 29st July in 2020, between 13:00 and 15:00 with a digital thermometer (GLA M800, GLA Agricultural Electronics, San Luis Obispo, CA). The THI values in the consecutive five-day ranges from 61~65 in the first of May and 78~82 in the late of July. Hence, the cows in May were not in heat stress, but the higher temperature in June and July resulted in THI values more than 72, so the cows were under heat stress. We have added the information in our revised manuscript, please see in Line 102-117.

  1. Lines 83–84 The phrase“first lactating period per lactation”is confusing. Please rephrase to clearly indicate whether you mean first lactation or a specific period within the current lactation.

Response: Thank you for your question. We mean that the middle of their first lactation period, and have revised in the manuscript Line 99-102.

Since the cows are in the process of maintaining peek milk production in the middle of their first lactation period (122.2 ± 11.6 days), we selected 68 cows with similar age (28.1±0.4 months) and daily milk yield (39.1 ± 1.9 kg per day) during this period.

Lines 84–96: It is unclear how 68 cows and then 12 cows were used. Were 68 cows initially evaluated, and then a subset of 12 was used for the omics analyses? Please clarify the selection process and final sample size used in each analysis step.

Response: Thank you for your question. Yes, it is right that we initially evaluated the THI, milk yield and RT values for 68 cows, then we performed PCA analysis on 68 dairy cows using two variables: RT measured at the end of July and the change of milk yield between May and July. Then we selected 12 cows, the 6 cows which the PC1 value more than 1.3 and the 6 cows which PC1 value less than -1.2. Please see the details for the PCA results in our previous paper [3] with supplemental file 4.

  1. Li, R. et al. (2023) Identification of target genes and pathways related to heat tolerance in Chinese Holstein cows. Livestock Science 271.

Results

  1. Figures

The figures are difficult to read due to resolution and size. Please improve the quality and adjust formatting to make results more legible.

Response: Thanks for your suggestion. We have improved the quality of all the figures. Please see in the revised manuscript.

  1. Supplementary Data

One of the supplementary files includes p-values greater than 1.00, which is not possible. Please check for typographical errors and correct them.

Response: Thanks for your kindly reminder. We have carefully checked the supplementary file 3. There were 32 KEGG pathways, and we have revised in the manuscript.

Discussion

  1. Integration of findings. Currently, the proteins and metabolites are discussed separately, without a clear link to their biological implications. Please expand the discussion to explain how these molecular changes affect the physiology, production, and health of cows under heat stress. This would help connect the omics results to practical outcomes and enhance the relevance for dairy science and animal management.

Response: Thank you for your suggestion. We have reorganized the discussion part. We firstly discussed the proteins and metabolites separately, and then we combined analyzed the proteins and metabolites interaction to influence the physiology, production and health of cows under heat stress in the revised manuscript Section Discussion.

Conclusion

  1. Concluding remarks

The conclusion is adequate but could be strengthened by explicitly stating how the integrated proteomic–metabolomic approach contributes to better understanding of heat stress and its potential application in managing dairy cow health and productivity.

Response: Thank you for your suggestion. We have improved the Conclusion Section based on your suggestions. Please see in below or in the revised manuscript Line 502-514.

In summary, our study identified 29 differentially expressed proteins (DEPs) and 338 differential metabolites after comparing HS and HR conditions. The integrated analysis of 29 DEPs and 338 metabolites revealed four key pathways underlying protein-metabolite interactions, where PLOD1, EXT1, GSN, and ACTN4 interacted with N6-Acetyl-L-lysine, citric acid, 4-Pyridoxic acid, uracil, uric acid, and arachidonic acid were enriched. Functional validation through cell experiments, qPCR, and Western blot analysis showed that interference of ACTN4 gene could induce dairy cow mammary epithelial cells apoptosis, which could be regarded as a potential biomarker for the selection of heat-tolerant cows. Overall, these findings enhanced our understanding of molecular mechanisms at proteomics and metabolomic levels by which dairy cows respond to HS condition, facilitate novel insights that may inform strategies to mitigate stress using the potential applications e.g., for targeted breeding strategies or nutritional interventions, and enhance animal welfare and productivity under high-temperature conditions.

Reviewer 2 Report

Comments and Suggestions for Authors

Dear, authors,

follow considerations about the paper.

Title: 

Abstract: to reduce the introduction of the summary, maxime 2 lines. Address more methodoly and results in this section. There is no methodology information in this section.

Line 28-28: What impacts do these results have on cow health? Describe them in this section.

Line 30-31: Is there a specific strategy to mitigate stress? What is the relationship with the results obtained? The conclusion is very broad; it needs to specify the contribution of the results of this study. Rephrase.

Introduction: The introduction is very complete and supported by an updated theoretical framework.

Materials and Methods:  

Line 91-93: This part is the result. Please enter it in the results section.

Line 96 -97: Was this study approved by the animal ethics and use committee? 

Line 111-128: According to which methodology?

Line 129-140: According to which methodology?

Line 228-234:  The PCA analysis cited in the methodology? Describe it here.

Results:

Figures needs to be improved in resolution for interpretation. 

Line 422-423: Why was lysine degradation the most important pathway? Justify this result.

Line 436-438: Does this explanation justify the question above? If so, please reference the above description in this paragraph.

The discussion is well founded and justifies the results obtained.

Kind Regards.

Author Response

Dear Reviewers:

We are really appreciating for your comments and suggestions for our manuscript entitled “Integrated analysis of proteomics and metabolomics for heat stress in Chinese Holstein cows (animals-3878666)”. Those comments are all very valuable and very helpful for revising and improving our manuscript. We have carefully considered the comments and made the necessary corrections, which we hope will meet with your approval. The details of the revised sections are marked in red and highlighted in yellow in the manuscript-marked. The main corrections in the paper and the responds to the editors and reviewer’s comments point by point are as flowing:

Abstract: to reduce the introduction of the summary, maxime 2 lines. Address more methodology and results in this section. There is no methodology information in this section.

Response: Thank you for your suggestion. We have reduced the introduction of the summary and addressed more methodology and results in this section. Please see in Line 31-35

Heat stress (HS) severely reduces milk yield and significantly causes substantial economic losses of dairy cows. This study aims to generate a more comprehensive understanding for HS regulation mechanisms, so TMT-based proteomes and untargeted metabolomics approach were used to conduct the integrated analysis of proteomics and metabolomics in heat-stressed (HS, n = 6) and heat-resistant (HR, n = 6) Chinese Holstein cows.

Line 28-28: What impacts do these results have on cow health? Describe them in this section.

Response: Thank you for your suggestion. The identified up-regulated PLOD1 and ACTN4 and down-regulated EXT1 and GSN interacting with the down-regulated N6-Acetyl-L-lysine, the citric acid, the 4-Pyridoxic acid, the uracil, and the uric acid and the up-regulated arachidonic acid could be used for rapid and noninvasive screening of heat-tolerant cows. Please see in Line 35-45.

The proteomics analysis showed that 29 differentially expressed proteins (DEPs) with SERPINA3-7 and ACTN4 and PLOD1 upregulated and GSN downregulated in HR cows. The metabolomics analysis showed that 168 differential positive metabolites, and 170 differential negative metabolites were identified with HR cows exhibiting lower levels of anti-inflammatory compounds such as N6-Acetyl-L-lysine. In addition, integrated analysis of 29 DEPs and 338 differential metabolites revealed four key pathways including lysine degradation (ko00310), metabolic pathway (ko01100), Fc gamma R-mediated phagocytosis (ko04666), and Amoebiasis (ko05146) underlying protein-metabolite interactions, where the up-regulated PLOD1 and ACTN4 and the down-regulated EXT1 and GSN interacting with the down-regulated N6-Acetyl-L-lysine, citric acid, 4-Pyridoxic acid, uracil, and uric acid and the up-regulated arachidonic acid were enriched, which could be used for rapid and noninvasive screening of heat-tolerant dairy cows.

Line 30-31: Is there a specific strategy to mitigate stress? What is the relationship with the results obtained? The conclusion is very broad; it needs to specify the contribution of the results of this study. Rephrase.

Response: Thank you for your suggestion. We have rephrased this conclusion.

Our results facilitate a better understanding of the molecular mechanism for HS trait of dairy cows and provide a crucial insight into the alternative strategies to enhance animal welfare and productivity under high-temperature conditions.

Introduction: The introduction is very complete and supported by an updated theoretical framework.

Response: Thank you for your evaluation.

Materials and Methods:

Line 91-93: This part is the result. Please enter it in the results section.

Response: We totally agree with your opinion, but these results have been published in our previous study, and this study we focus on the proteomics and metabolomics analysis to reveal molecular mechanism for the heat stress in Chinese Holstein cows. Thus, this study is based on our previous study, and is an extension of our previous study [1]. Please see our previous study ‘Li, R. et al. (2023) Identification of target genes and pathways related to heat tolerance in Chinese Holstein cows. Livestock Science 271’. Hence, we put the sentence in the Materials and Methods section.

Line 96 -97: Was this study approved by the animal ethics and use committee?

Response: Yes, the study was approved by the animal ethics in the Ethical Approval section. Please see in the revised manuscript Line 545-547

Ethical Approval: The animal study was reviewed and approved by the Huazhong Agriculture University Animal Management and Ethics Committee (HZAUCA-2019-002). All methods were carried out in accordance with relevant guidelines and regulations.

Line 111-128: According to which methodology?

Response: Thank you for your question. I guess you would like to know which method we are according for the FASP. We used the method from the paper “Universal sample preparation method for proteome analysis”, we have cited it in the manuscript.

Line 129-140: According to which methodology?

Response: Thank you for your question. I guess you would like to know which method we are according for the TMT, we use a tandem mass tag (TMT) reagent according to the manufacturer’s instructions (Thermo Fisher Scientific Inc.).

Line 228-234: The PCA analysis cited in the methodology? Describe it here.

Response: Thank you for your question. As we have detailed the PCA analysis in our previous study [1] Li, R. et al. (2023) Identification of target genes and pathways related to heat tolerance in Chinese Holstein cows. Livestock Science 271. In this study, the samples we selected was same as our previous study. Hence, it is better not to repeat the same thing in this paper.

  1. Li, R. et al. (2023) Identification of target genes and pathways related to heat tolerance in Chinese Holstein cows. Livestock Science 271.
  2. Minamino, T. et al. (2010) Endoplasmic reticulum stress as a therapeutic target in cardiovascular disease. Circulation research 107 (9), 1071-1082.
  3. Qi, Y. and Xu, R. (2018) Roles of PLODs in collagen synthesis and cancer progression. Frontiers in cell and developmental biology 6, 66.
  4. Yuan, B. et al. (2022) PLOD1 acts as a tumor promoter in glioma via activation of the HSF1 signaling pathway. Molecular and Cellular Biochemistry, 1-9.

Results: Figures need to be improved in resolution for interpretation.

Response: Thanks for your suggestion. We have improved the quality of all the figures. Please see in the revised manuscript, and we also provided original figures with .tiff format when we resubmitted the manuscript.

Line 422-423: Why was lysine degradation the most important pathway? Justify this result. Line 436-438: Does this explanation justify the question above? If so, please reference the above description in this paragraph.

Response: Thanks for your question. We would like to answer these two questions together. The lysine degradation was the important pathway because of the lysine degradation where PLOD1-N6-Acetyl-L-lysine interaction was enriched (Table 3). PLOD1 plays the essential role in protein synthesis, folding, and trafficking and par-ticipates in fatty acid metabolism and collagen synthesis. in addition, the PLOD1 could enhanced the transcriptional activity of heat shock factor 1 (HSF1). Hence, HSF1 is activated during thermal stress may be caused by the transcriptional activation of PLOD1. Please find in Line 486-485.

The lysine degradation where PLOD1-N6-Acetyl-L-lysine interaction was enriched (Table 3). PLOD1 plays an essential role in protein synthesis, folding, and trafficking and participates in fatty acid metabolism and collagen synthesis [2, 3]. In addition, the PLOD1 could enhance the transcriptional activity of heat shock factor 1 (HSF1) [4]. Hence, HSF1 is activated during thermal stress may be caused by the transcriptional activation of PLOD1 [4].

The discussion is well founded and justifies the results obtained.

Response: Thank you for you evaluation

Kind Regards.

The authors.

Reviewer 3 Report

Comments and Suggestions for Authors

Please check the attached comment files.

Author Response

Dear Reviewers:

We are really appreciating for your comments and suggestions for our manuscript entitled “Integrated analysis of proteomics and metabolomics for heat stress in Chinese Holstein cows (animals-3878666)”. Those comments are all very valuable and very helpful for revising and improving our manuscript. We have carefully considered the comments and made the corrections, which we hope will meet with your approval. The details of the revised sections are marked in red and highlighted in yellow in the manuscript-marked. The main corrections in the paper and the responds to your comments point by point are as flowing:

General comment:

This study presents an integrated analysis of heat stress in dairy cattle. The results suggest that taking a comprehensive approach that encompasses both proteomics and metabolomics could improve our understanding of the physiological mechanisms underlying heat stress in dairy cattle. These insights could inform the development of precision feeding and breeding strategies based on omics data. However, the study could be strengthened by providing a more detailed description of the experimental design and a more in-depth discussion of the results, and issues such as missing data and low figure resolution should be addressed. Several specific comments are provided below.

Specific comment:

  1. Abstract
  • 19-20: Please briefly indicate the number of animals and how they were grouped.

Response: We have briefly indicated the number of animals in HS and HR groups. Please see in the revised manuscript Line 32 to 35.

This study aims to generate a more comprehensive understanding for HS regulation mechanisms, so TMT-based proteomes and untargeted metabolomics approach were used to conduct the integrated analysis of proteomics and metabolomics in heat-stressed (HS, n = 6) and heat-resistant (HR, n = 6) Chinese Holstein cows.

  • 28-31: Instead of the final sentence, it would be more meaningful to highlight the biological significance of the identified pathways, particularly what they imply about heat stress resistance.

Response: Thank you for your suggestion. We have revised the abstract part. Please see in the advised manuscript Line 35-51.

The proteomics analysis showed that 29 differentially expressed proteins (DEPs) with SERPINA3-7 and ACTN4 and PLOD1 upregulated and GSN downregulated in HR cows. The metabolomics analysis showed that 168 differential positive metabolites, and 170 differential negative metabolites were identified with HR cows exhibiting lower levels of anti-inflammatory compounds such as N6-Acetyl-L-lysine. In addition, integrated analysis of 29 DEPs and 338 differential metabolites revealed four key pathways including lysine degradation (ko00310), metabolic pathway (ko01100), Fc gamma R-mediated phagocytosis (ko04666), and Amoebiasis (ko05146) underlying protein-metabolite interactions, where the up-regulated PLOD1 and ACTN4 and the down-regulated EXT1 and GSN interacting with the down-regulated N6-Acetyl-L-lysine, citric acid, 4-Pyridoxic acid, uracil, and uric acid and the up-regulated arachidonic acid were enriched, which could be used for rapid and noninvasive screening of heat-tolerant dairy cows. Functional validation through cell experiments, qPCR, and Western blot analysis showed that interference of ACTN4 gene could induce dairy cow mammary epithelial cells apoptosis, which could be regarded as a potential biomarker for HS issue in Chinese Holstein cows. Our results facilitate a better understanding of the molecular mechanism for HS trait of dairy cows and provide a crucial insight into the alternative strategies to enhance animal welfare and productivity under high-temperature conditions.

  1. Materials and Methods
    • This study appears to have used records from approximately 1,000 cows to select experimental animals. Please specify the source of these records and explain why this particular stage was chosen. Also, the ethics approval number for the animal experiment appears to be missing.

Response: Thank you for your question. We selected the 69 cow in the middle of their first lactation period (122.2 ± 11.6 days) with similar age (28.1±0.4 months) and daily milk yield (39.1 ± 1.9 kg per day) from 1000 lactating healthy Chinese Holstein dairy cows which housed in the same farm in Jinan City, China. The cows are in the process of maintaining their peak milk production in the middle of their first lactation period, and the primary focus during this stage is to maintain production for as long as possible. The management and the environment factors could help maintain the production. Heat stress is an important environment factors to influence milk production period, so we choose this particular stage to study. We have explained in the revised manuscript, please see in Line 99-102.

Since the cows are in the process of maintaining peek milk production in the middle of their first lactation period (122.2 ± 11.6 days), we selected 68 cows with similar age (28.1±0.4 months) and daily milk yield (39.1 ± 1.9 kg per day) during this period.

The ethics approval number for the animal experiment have been listed in Line 545-547. Please see the revised manuscript.

Ethical Approval: The animal study was reviewed and approved by the Huazhong Agriculture University Animal Management and Ethics Committee (HZAUCA-2019-002). All methods were carried out in accordance with relevant guidelines and regulations.

  • Out of 1,000 cows, 68 were selected and grouped according to rectal temperature and milk yield. These grouping results should be presented as data. Although such information seems to be described in Li et al., 2023, it would be helpful to briefly include the key information necessary to understand the present study.

Response: Thank you for your suggestion. We have detailed the main results from Li et al., 2023. Please see the revised manuscript in Line 99-118.

Since the cows are in the process of maintaining peek milk production in the middle of their first lactation period (122.2 ± 11.6 days), we selected 68 cows with similar age (28.1±0.4 months) and daily milk yield (39.1 ± 1.9 kg per day) during this period. They were all fed with the same total mixed rations and clean water to keep the same environmental conditions. We recorded separately the RT values for five consecutive days in 5st to 9st in May and 25st to 29st July in 2020, between 13:00 and 15:00 with a digital thermometer (GLA M800, GLA Agricultural Electronics, San Luis Obispo, CA). The ambient temperature (T) and relative humidity (RH) were also recorded to calculate the temperature-humidity index (THI) values. The THI values were calculated following the formula of THI = (1.8 × T + 32) - [(0.55 - 0.0055 × RH) × (1.8 × T - 26)] in the previous study [5, 21]. Based on our previous study, the THI range in the barn was 61-65 in early May which is not in a HS state, but its value is 78-82 in late July which was under heat stress [5]. In addition, we used principal component analysis (PCA) for the RT and milk yield of these 68 cows. The first principal component (PC1) explained the 58% variation of the change in RT and milk yield due to HS. According to PC1, 12 Chinese Holstein cows were selected where 6 cows were assigned to the HS group (PC1 > 1.3) and 6 cows were assigned to the heat resistance (HR) group (PC1 < -1.2). The average RT and milk yield for 5 days between the two groups exhibited a significant difference in late July [5]. More detailed information for grouping the HS and HR cows using the PC1 values has been carefully described in our previous study [5].

  • Please specify when the blood samples were collected.

Response: The blood samples were collected in 31st July after we have grouped the samples. Please see the revised manuscript Line 119-121.

For the plasma samples collected from the tail veins of each individual of the 12 selected cows in the end of July, one 10 ml vacutainer blood collection tube containing the EDTA-coated RNase-free was used.

  • 104: Please provide the full name of the SDT buffer.

Response: Thank you for your suggestion. The full name of the SDT buffer is SDT lysis buffer, which is a mix agent which usually contains 2% Sodium dodecyl sulfate (SDS), 100mM Dithiothreitol (DTT) and 100mM Tris-HCl.

We added the SDT Lysis buffer (2% Sodium dodecyl sulfate (SDS), 100mM Dithiothreitol (DTT) and 100mM Tris-HCl), boiled for 15 min, and then centrifuged at 14,000 × g for 40 min.

  • 125: 7%-40%? It seems to contain an error. Please verify.

Response: Thank you for pointing out this error. We have deleted 7-40%. And we added the liquid phase gradient of chromatographic column in Table S2. Please see in revised manuscript Line 149-151.

Table. S2 Liquid phase gradient of chromatographic column

Time(min

Buffer B gradient

0~25

0

25~30

0%~7%

30~65

7%~40%

65~70

40%~100%

70~85

1

and buffer B (10 mM HCOONH4, 85% acetonitrile, pH = 10.0) for 30-65 min at a flow rate of 1 mL/min with a phase gradient (TableS2).

  • 195 Please spell out the full name of ACTN4 at first mention.

Response: Thank you, we have revised it.

  • 197: Please clarify why human embryonic kidney cells were used in this study.

Response: The human embryonic kidney (HEK293) cells were used in this study to verify the interference efficiency for the shRNAs of ACTN4 gene. HEK293 cells are very easy to be transfected and can efficiently express foreign genes.

  • Although references are cited, basic details of qPCR and Western blot analysis (e.g., primer information, cycling conditions, antibody details, and detection method) should be provided to ensure reproducibility.

Response: The primers have been detailed in Table S2. The antibody details have been detailed also. In addition, we used ImageJ software to calculate the integrated density for the western blot band. Please see in the revised manuscript Line 240-244. The statical results for integrated density have been showed in Figure S5C and Figure S5D. We also have cited the Figures in the revised manuscript Line 295.

Table S2. The primers of qRT-PCR for genes related to heat tolerance and the designed siRNAs for actinin alpha 4 gene (ACTN4) gene.

Primer name

Primer sequence5’-3’

ACTN4-F

CCCGACGAGAAGGCCATAAT

ACTN4-R

TGCTCGTTCTCCTGGTTGAC

HSP70-F

AGAAGAAGGTGCTGGACAAGT

HSP70-R

CTGGTACAGTCTGCTGATGATG

HSP27-F

CGTCAAGGTGGTGGACAAC

HSP27-R

TCGGATGAGACAGTGGACAC

caspase3-F

GAACTGGACTGTGGTATTGAGAC

caspase3-R

CGAGCCTGTGAGCGTACTT

Cytc-F

AGTGTGCCCAGTGCCATAC

Cytc-R

CTCCATCAGCGTCTCCTCTC

Bcl-2-F

TTCGCCGAGATGTCCAGTC

Bcl-2-R

GGTTGACGCTCTCCACACA

Bax-F

CGGAGATGAATTGGACAGTAACA

Bax-R

CAGTTGAAGTTGCCGTCAGAA

GAPDH-F

GAAGGTCGGAGTGAACGGAT

GAPDH-R

TTCTCTGCCTTGACTGTGCC

sh-1-ACTN4

GCAAGATGGTGTCGGACATCA

sh-2-ACTN4

GGAAGGATGGTCTTGCCTTCA

sh-3-ACTN4

GCAGTATGAGCGCAGCATTGT

NC

TTCTCCGAACGTGTCACGT

HSPH1 Antibody (66723-1-IG, Invitrogen) at dilution of 1:10000 and LBP Antibody (RR433-8, Invitrogen) at dilution of 1:50 incubated at room temperature for 1.5 hours. ImageJ software was used to calculate the integrated density for the band, and each band was measured three times. The student’s t-test was used to calculate the significant difference between two groups.

Figure S5. The western blot results of (A) HSPH1 and (B) LBP proteins. The statistical analysis for the integrated density of (C) HSPH1 and (D) LBP western blot result.

  1. Results
    • 250: Please confirm whether the sample size is 12. I think the data suggest 4 vs 4 comparison. Please indicate the number of samples in the table footnotes. I am concerned whether comparing only four animals per group is sufficient to generalize the findings to heat resistant and heat susceptible groups.

Response: Thank you for the question. We totally used 8 samples in HS and HR group; each has 4 samples, for mass spectrometry analysis of proteomics sequencing. However, we used 6 samples in HS and HR group for LC-MS analysis of metabolomes sequence. We have clearly stated that in the footnotes of Table 1 and in the footnotes of Figure 5 and Figure 6. We also clearly stated the samples number in the Materials and Methods section 2.3 and section 2.6. Please see in the revised manuscript.

  • 262: Please clarify whether the p-values reported here are adjusted. Since the statistical methods section indicates that adjusted p-values were calculated, it would be clearer to explicitly state “adjusted p-value” in the text.

Response: Thank you for your suggestion. We have made sure that the statistical method is adjusted p-value, and have been stated in the revised manuscript.

  • Table 1: Please consider adding fold-change values for each protein and specifying in the table footnote that the p-values are adjusted.

Response: Thanks for your suggestion. We have added the HR_HO/HS_HO in Table 1, and added the footnote that the P-values are adjusted. Please see in the revised manuscript.

Table 1. The top 10 differentially expressed proteins (DEPs).

UniProKB

GeneName

AAs

Coverage [%]

Peptides

PSMs

Unique Peptides

AAs

MW

calc.

Score Mascot:

Regulation

P-value

HR_HO/HS_HO

[kDa]

 pI

Mascot

A2I7N3

SERPINA3-7

417

49

22

394

13

417

46.9

6.01

6896

UP

0.0043325

2.29

A5D7D1

ACTN4

1032

1

1

1

1

1032

117

6.38

0

UP

0.0047489

1.76

Q3ZEJ6

SERPINA3-3

411

43

15

326

5

411

46.1

6.48

8029

UP

0.0011647

1.66

A2I7N1

SERPINA3-7

415

33

16

448

1

415

46.5

5.81

10012

UP

0.0020006

1.61

Q0VCQ9

RCN2

317

3

1

1

1

317

36.9

4.4

0

UP

0.0009764

1.53

F6PYF1

MMP19

499

3

1

1

1

499

56.4

6.77

42

UP

0.0100722

1.50

UPI0000616416

MGC137014

191

66

10

113

10

191

20.6

8.59

2185

DOWN

0.0142715

0.83

A0A4W2EXA9

THSD7A

1676

1

1

2

1

1676

186.8

7.43

0

DOWN

0.0083202

0.83

UPI0000693DEA

212

50

9

67

9

212

22.5

7.46

1662

DOWN

0.0074206

0.81

A5D7I4

EXT1

746

2

1

1

1

746

86.2

9.04

30

DOWN

0.0191447

0.75

F1N1I6

GSN

846

61

42

316

1

846

92

6.86

11356

DOWN

0.0134129

0.71

Note: P-values are adjusted. Each groups have 4 samples.

  • All Figure: The figure resolution appears low (possibly due to my computer), which makes it difficult to evaluate the data accurately. Please provide high resolution versions or the original files for proper assessment.

Response: Thanks for your suggestion. We have improved the quality of all the figures. Please see in the revised manuscript, and we also provided original figures with .tiff format when we resubmitted the manuscript.

  • Figure 7: Data do not seem to be available. Please check and confirm. It is difficult to evaluate the paragraph in Section 3.8. L.401: Why was only ACTN4 analyzed in detail? The rationale is unclear.

Response: Thank you for your suggestion. It is better to answer these two questions together. As we have selected the upregulated ACTN4 in HR group, which could be regard as a biomarker for the heat stress cow selection. However, we first need to verify gene’s function to milk production. Milk production is highly dependent on the optimal development of the mammary epithelium. It is therefore essential to better understand the functions of ACTN4 to mammary epithelial cell growth. We need more work to verify the genes’ function to mammary epithelial cell growth, and we will continue our work for verifying the function of PLOD1, EXT1, and GSN at cell level.

  1. Discussion and Conclusions

I consider the limited interpretation of the data in the discussion and conclusion sections to be a weakness of the study. One of the major strengths of this work, in my opinion, is its integrated omics approach combined with functional validation through cell experiments, qPCR and Western blot analyses. However, the manuscript does not present a conclusion that clearly explains the biological significance of these results from an integrative perspective.

While identifying significantly altered pathways between groups is important, further elaboration on the physiological implications of these findings in the heat-resistant group and the potential insights they offer would be beneficial. Linking the identified pathways to metabolic adaptations, protein synthesis regulation or other biological processes under heat stress, and explaining how these responses are linked to the traits of the heat-resistant group, would clarify and enhance the conclusion.

Response: Thank you for your suggestion. We have reorganized the discussion part. We firstly discussed the proteins and metabolites separately, and then we combined analyzed the proteins and metabolites interaction to influence the physiology, production and health of cows under heat stress in the revised manuscript Section Discussion Line 445-496. We also improved the Conclusion Section based on your suggestions. Please see in below or in the revised manuscript Line 497-510.

  1. Discussion

In this study, the plasma metabolome and proteome were integrated to uncover the molecular regulation of HS to physiology, production and health in Chinese Holstein dairy cows. Plasma proteomics and metabolomics between HS and HR cows identified 29 proteins and 338 metabolites in the differentially expressed conditions.

Four cytoplasmic vesicle-localized serpin proteins were significantly up-regulated in the HR group, including serpin A3-3, serpin A3-5, serpin A3-7, and serpin A3-8, which modulate the immune responses by targeting proteases involved in HS regulation [37-39] . In addition, consistent with our previous study that heat shock protein (HSPH1) was up-regulated in the HS group, it was divergently expressed in indigenous and crossbreed cattle [37] and was highly expressed in the blood of Zebu cattle [40] and the liver of lactating dairy cows after heat exposure. We also found that HSPH1 protein is connected with HSPA8 [41] and HSP90B1 [42], which are critical for coordinating the biological system changes under HS conditions.

The differential metabolites of N6-Acetyl-L-lysine, citric acid, 4-pyridoxic acid, uracil, uric acid, and arachidonic acid interacting with four DEPs were mainly involved in the primary glomerulopathies, COVID-19, Alzheimer’s disease, obesity, and metabolic dysfunction-associated steatotic liver diseases [43-46]. Nα-acetyl-α-lysine plays a role in HS responses in various organisms. It can accumulate and act as a probable thermolyte to alleviate HS after increasing in concentration under elevated temperatures [47]. The application of citric acid may alleviate growth and physiological damage caused by high temperature, as exogenous citric acid enables to maintain membrane stability and root activity. Meanwhile, citric acid can activate HSP genes and antioxidant response to protect tall fescue against HS-induced damage in the plant study [48]. As a metabolite of vitamin B6, 4-Pyridoxic acid has been observed to increase under HS conditions, together with inflammatory and oxidative stress in various studies. It suggested the disruption role of HS on the body's normal vitamin B6 metabolism [49]. As an intermediate breakdown product of guanosine monophosphate (GMP) and uracil, dihydrouracil is decreased probably because of the defects of human metabolism. Of note, GMP production is mediated by CD39 and CD73 and their axis plays a crucial role in immunity and inflammation [50, 51]. Uric acid is generated during HS partly due to nucleotide release from muscles. The HS and high uric acid levels are linked, especially when the individuals are exposed to the heating conditions with inadequate hydration [52].

Four DEPs (i.e. up-regulated PLOD1 and ACTN4 and down-regulated EXT1 and GSN) interacting with six differential metabolites (i.e., the down-regulated N6-Acetyl-L-lysine, the citric acid, the 4-Pyridoxic acid, the uracil, and the uric acid and the up-regulated arachidonic acid) were enriched in four key pathways of underlying protein-metabolite interactions. The lysine degradation where PLOD1-N6-Acetyl-L-lysine interaction was enriched (Table 3). PLOD1 plays an essential role in protein synthesis, folding, and trafficking and participates in fatty acid metabolism and collagen synthesis [53, 54]. In addition, the PLOD1 could enhance the transcriptional activity of heat shock factor 1 (HSF1) [55]. Hence, HSF1 is activated during thermal stress may be caused by the transcriptional activation of PLOD1 [55]. More interesting, the amoebiasis where ACTN4-arachidonic acid interaction was enriched. ACTN4 protein is an important structural protein for cytoskeleton organization and maintenance of cellular integrity, which also plays an important role in adipogenesis and marbling deposition in cattle [56, 57]. The release of arachidonic acid increased after HS, to trigger autophagy in sertoli cells via dysfunctional mitochondrial respiratory chain function, which potentially affect cellular function and reproductive health [58]. In addition, previous researchers also found that arachidonic acid had high sensitivity and specificity in diagnosing HS status [59], which could be regarded as one of the potential biomarkers for HS in dairy cows. However, the molecular relationship between PLOD1 and HSF1 or ACTN4 and arachidonic acid to HS needs more experimental verification in dairy cows in the further study.

  1. Conclusions

In summary, our study identified 29 differentially expressed proteins (DEPs) and 338 differential metabolites after comparing HS and HR conditions. The integrated analysis of 29 DEPs and 338 metabolites revealed four key pathways underlying protein-metabolite interactions, where PLOD1, EXT1, GSN, and ACTN4 interacted with N6-Acetyl-L-lysine, citric acid, 4-Pyridoxic acid, uracil, uric acid, and arachidonic acid were enriched. Functional validation through cell experiments, qPCR, and Western blot analysis showed that interference of ACTN4 gene could induce dairy cow mammary epithelial cells apoptosis, which could be regarded as a potential biomarker for the selection of heat-tolerant cows. Overall, these findings enhanced our understanding of molecular mechanisms at proteomics and metabolomic levels by which dairy cows respond to HS condition, facilitate novel insights that may inform strategies to mitigate stress using the potential applications e.g., for targeted breeding strategies or nutritional interventions, and enhance animal welfare and productivity under high-temperature conditions.

Round 2

Reviewer 1 Report

Comments and Suggestions for Authors

I have carefully reviewed the revised version of your manuscript and I am very pleased with the improvements you have made. The additional explanations and clarifications have significantly enhanced the clarity, coherence, and overall quality of the paper. Your work addresses a highly relevant and timely topic and represents a valuable contribution to the field. I appreciate the thoughtful effort you put into revising the manuscript—congratulations on an excellent and impactful piece of research!

Reviewer 3 Report

Comments and Suggestions for Authors

The authors have appropriately revised the manuscript according to the comments, and the key issues raised in the review appear to have been adequately addressed.